# Divergent evolution of sleep in *Drosophila* species

Michaela Joyce[1,2], Federica A. Falconio[1,4], Laurence Blackhurst[1,4], Lucia Prieto-Godino [2], Alice S. French [1,2,3] ✉ & Giorgio F. Gilestro [1] ✉

Living organisms synchronize their biological activities with the earth's rotation through the circadian clock, a molecular mechanism that regulates biology and behavior daily. This synchronization factually maximizes positive activities (e.g., social interactions, feeding) during safe periods, and minimizes exposure to dangers (e.g., predation, darkness) typically at night. Beyond basic circadian regulation, some behaviors like sleep have an additional layer of homeostatic control, ensuring those essential activities are fulfilled. While sleep is predominantly governed by the circadian clock, a secondary homeostatic regulator, though not well-understood, ensures adherence to necessary sleep amounts and hints at a fundamental biological function of sleep beyond simple energy conservation and safety. Here we explore sleep regulation across seven *Drosophila* species with diverse ecological niches, revealing that while circadian-driven sleep aspects are consistent, homeostatic regulation varies significantly. The findings suggest that in Drosophilids, sleep evolved primarily for circadian purposes. The more complex, homeostatically regulated functions of sleep appear to have evolved independently in a species-specific manner, and are not universally conserved. This laboratory model may reproduce and recapitulate primordial sleep evolution.

Studying the evolution of sleep is paramount to understand its functions, and in the past century extensive work has gone to characterize sleep in the most disparate species[1,2], often unveiling puzzling findings. We still do not understand, for instance, how some animals—such as elephants[3], giraffes[4] or cavefish[5]—spontaneously sleep less than 1 or 2 h a day, while others—like bats[6] or koalas[7]—almost completely fill their days with sleep; yet others, such as migratory birds, can adapt their sleep amount to their changing ecological needs, compressing it from many hours to mere minutes a day when the migratory instinct commands[8,9]. The disparities observed in sleep patterns among various species can be largely attributed to differences in sleep homeostasis rather than circadian control, hinting at an adaptive response to environmental pressures and ecological demands, which can vary drastically even within closely related species. The variability in homeostatic sleep regulation implies a level of evolutionary flexibility in how sleep is prioritized and managed, potentially reflecting differing survival strategies or metabolic requirements. For example, animals like migratory birds adjust their sleep drastically, through an unknown biological mechanism that allows for significant modulation of sleep needs in response to ecological demands. This variability has profound implications for our understanding of sleep's fundamental purposes in animals and humans alike, suggesting that while the circadian control of sleep might be relatively conserved, the homeostatic regulation of sleep is highly adaptable and evolved in response to specific environmental challenges and lifestyle constraints. Most of the observations on the evolutionary variety of sleep and its plastic adaptation are important and insightful[5], but conclusions are limited by the large evolutionary distance of the studied species. Here, we introduce the *Drosophila* genus as an ideal

[1]Department of Life Sciences, Imperial College London, London, UK. [2]The Francis Crick Research Institute, London, UK. [3]School of Physiology, Pharmacology and Neuroscience, University of Bristol, Bristol, UK. [4]These authors contributed equally: Federica A. Falconio, Laurence Blackhurst. ✉e-mail: alice.french@bristol.ac.uk; giorgio@gilest.ro

evolutionary playground to study the behavioral and genetic bases of sleep traits through a genetically well characterized group of animals at divergent evolutionary distances spanning 50+ million years of evolution. We show that the spontaneous circadian-driven aspects of sleep are conserved among all species but the homeostatic regulation, unexpectedly, is not. We uncover differences in the behavioral, cell-biological and neuro-pharmacological aspects of sleep and suggest that, in Drosophilids, sleep primarily evolved to satisfy a circadian role, keeping animals immobile during dangerous hours of the day.

## Results

### Diverse sleep patterns across *Drosophila* species

This study focuses on seven species with divergent ecological niches[10] (Fig. 1a) and geographical ancestral origins[11–14] (Fig. 1b). Using an automatic video tracking system[15], we analyzed spontaneous sleep in multiple independently-caught natural strains and found a qualitatively similar pattern of sleep and activity in all the species analyzed (Fig. 1c and Supplementary Fig. 1a, b). In oscillating light-dark conditions, all Drosophilids showed recognizable crepuscular peaks of activity (Supplementary Fig. 1a, b), with their sleep mostly concentrated during the night and with male animals showing a prominent post-meridian sleep episode (Fig. 1c, e), previously identified in *D. melanogaster* as "*siesta*"[16] and tightly bound to the circadian clock[17]. In terms of circadian regulation of sleep and activity, *D. virilis* was the only outlier in our group, showing no sexual dimorphism in the *siesta* (Fig. 1c, e) and—as previously reported[18]—strong arrhythmicity in the absence of circadian entrainment by light (Fig. 1d). *D. virilis* is currently considered a cosmopolitan species, but its widespread localization is believed to be a very recent development in evolutionary terms, most likely linked to human movements. *D. virilis* may have originated from and adapted to the arid regions of Iran or Afghanistan from the early Miocene[19] and this evolutionary selection may explain the unusual display of afternoon *siesta* in females (Fig. 1c–e), likely evolved as a sheltering mechanism to escape the hostile conditions of a desertic afternoon. A similar phenotype was recently observed in the closely related *D. mojavensis*, and *D. arizonae* for which it was also proposed to be a mechanism of evolutionary adaption to stressful desertic conditions[20].

### Conservation of sleep depth in Drosophilids: a novel paradigm

To investigate how profound is the evolutionary conservation of sleep in Drosophilids, we created a novel paradigm for the analysis of sleep depth. The system builds[21] on a previously described hidden Markov chain (HMC) model[22], here complemented and validated by a robotic system able to measure arousal threshold by challenging flies with air puffs. These "challenging stimuli" are delivered after random periods of total or partial inactivity (ranging from 1 to 60 min; Fig. 1f–h and Supplementary Fig. 1d, e) and at the end of each experiment, response data are analyzed *ex post*[21] to determine whether behavioral bouts that are differently labeled by the HMC model are also characterized by a different arousal threshold. Strikingly, the validation proved this to be the case (Fig. 1h). Bouts that were labeled as "light sleep" by the model also happened to have a strong probability of response to challenging stimuli, while bouts labeled as "deep sleep" did not. Unsurprisingly, the strongest probability of response was observed when the stimuli were delivered during bouts that the model would have labeled as "wakefulness" (Fig. 1h). Overall, the combined analysis of sleep staging and arousal threshold proved crucial in three ways: (1) it confirmed that the ethoscope based video tracking system could indeed reliably detect sleep in all species, even though it was initially developed for *D. melanogaster*[15]; (2) it liberated our analysis from the arbitrary definition of sleep as 5 min of immobility (also known as the 5 min rule[22–24]) by adopting an agnostic criterion instead; (3) it allowed us

to quantify and describe the actual composition of different sleep stages during the 24 h, as well as their difference across species (Supplementary Figs. 1 and 2). In particular, the analysis showed that all species experience more light sleep than deep sleep, with an average ratio of $1.5 \pm 0.4$ (mean $\pm$ SD; excluding the outlier *D. erecta* that has a ratio of 10.2). In all species, deep sleep mostly concentrates in the early hours of the night (ZT 12–15–Supplementary Fig. 1d) and, exception made for *D. erecta* and *D. virilis*, the early night is also the only moment during which flies are more likely to experience deep rather than light sleep. The behavioral bouts identified by the model different by length, with deep sleep and active wake being considerably longer than, respectively, light sleep and quiet wake (Supplementary Fig. 2). However, length is not the sole determinant, given that many episodes of light or deep sleep share the same duration and must therefore be disambiguated on the basis of the model's transition probability matrix (Fig. 1g for *D. melanogaster*, and Supplementary Table 1 for all other species). While providing the unique ability of separating light sleep from deep sleep, the HMC analysis also confirms that the original 5 min rule can still be used to disambiguate sleep from wakefulness given that, in all species, 100% of the bouts of inactivity lasting five or more minutes are classified as sleep by the model. In non-melanogaster species, 100% of the deep sleep episodes feature at least 8 min of consecutive inactivity (Supplementary Fig. 2b).

### Homeostatic sleep regulation: species-specific responses to deprivation

It appears, therefore, that those aspects of sleep and activity that are meant to be regulated by the circadian clock are generally well conserved among Drosophilids but what about the second layer of regulation, homeostasis? In any animal, the easiest way to measure sleep homeostasis is to forcefully reduce sleep amount through techniques of mechanical interference to subsequently induce, and then quantify, a period of recovery sleep known as "rebound sleep". To test rebound sleep across species we combined ethoscopes with a robotic machine able to selectively disturb the sleep of single flies by rotating their housing tube after every 30 s of immobility[15,25] (Fig. 2a). To limit the amount of stress and confounding factors, the machine will interfere exclusively with flies that are immobile for a specified amount of time, leaving the awake experimental companions undisturbed[15]. We found all seven tested species to be sensitive to this kind of mechanical interference, although with different degrees of arousal threshold (Supplementary Fig. 3a). We then subjected all species to the most efficient stimulus (400 rpm, 3 s) for 24 h (Fig. 2a) thus forcing all the treated animals to lose most, if not all, of their sleep for the duration of the experiments (Supplementary Fig. 3b–d). Surprisingly, only *D. melanogaster* showed signs of sleep rebound after mechanical sleep deprivation and no rebound was detected in any of the other species, neither in terms of sheer sleep amount (Fig. 2a), nor in terms of sleep depth (Fig. 2c). After sleep deprivation, *D. melanogaster* slept longer (and lab raised CantonS *D. melanogaster* slept deeper too–Fig. 2c) but all other species showed no signs of homeostatic rebound. Even more impressively, this difference between species persisted when sleep deprivation was pushed further to cover an uninterrupted period of 168 h: after 7 full days of continuous mechanical sleep deprivation, *D. melanogaster* showed a homeostatic rebound as previously reported[25], but none of the other six Drosophilae did (Supplementary Fig. 4). The technical nature of this experiment also allowed us to infer and compare the build-up in sleep pressure, which, in conditions of chronic deprivation, is a manifestation of homeostasis. Given that our robotic ethoscopes are programmed to deliver mechanical stimuli (tube rotations) only to animals showing signs of sleep, the number of delivered stimuli over time can be used as a proxy of sleep pressure: the stronger the desire to rest, the higher

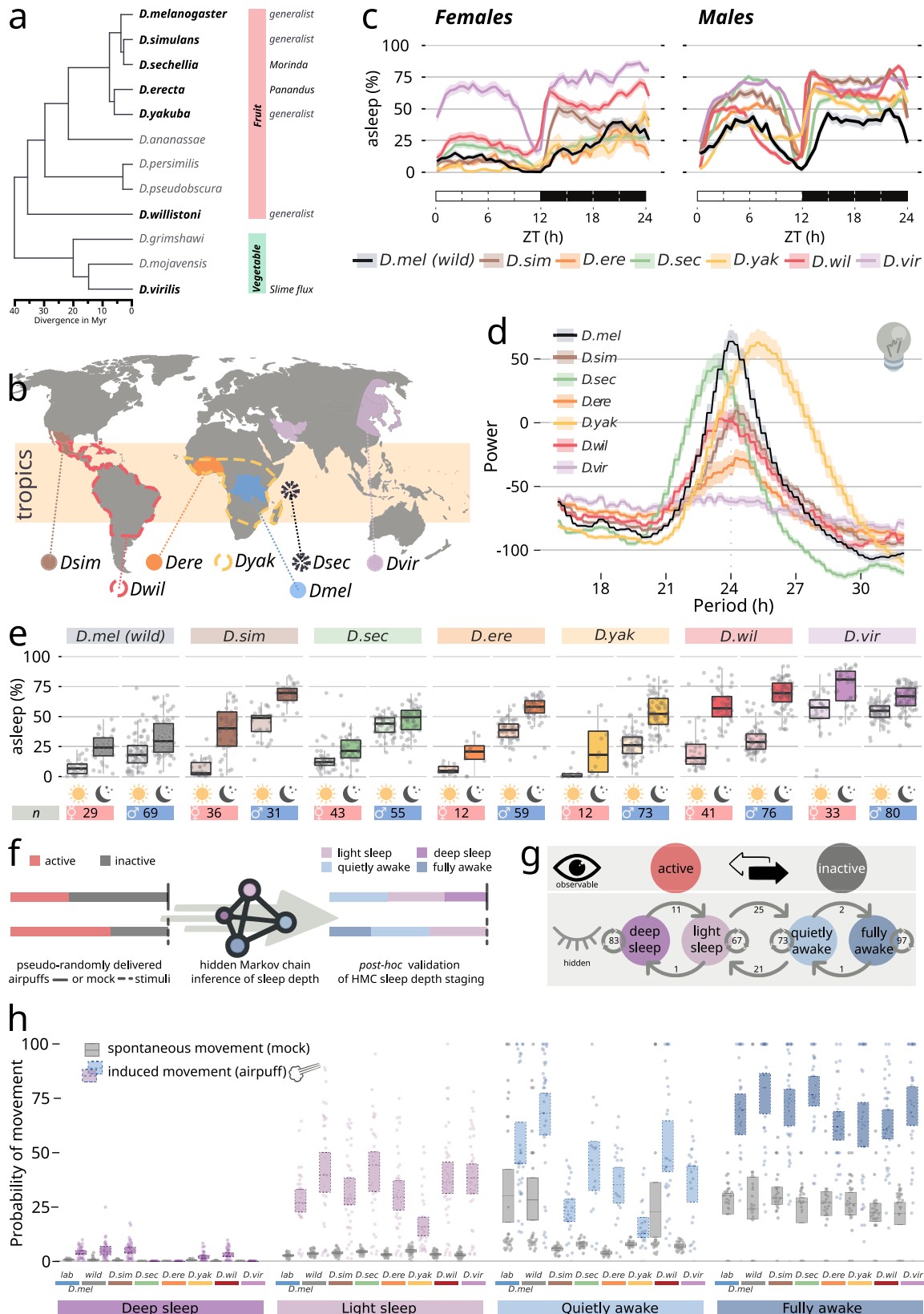

the number of stimuli received[25]. A significant increase in sleep pressure over a week was noted exclusively in *D. melanogaster* (Supplementary Fig. 5). In contrast, no other species demonstrated an increase in sleep pressure during the 7 days of sleep deprivation and, if anything, almost all of them showed a decrease in sleep pressure: perhaps a sign of plasticity and adaptation to the sleep-

deprivation treatment (Supplementary Fig. 5). Notably, none of the species analyzed showed any sign of lethality after 1 week of sleep deprivation (Supplementary Fig. 6), in full accordance with earlier findings in *D. melanogaster* indicating that prolonged lack of sleep has no automatic consequences on longevity[25–27]. Finally, no sleep rebound could be observed even when flies never experienced light

**Fig. 1 | Spontaneous sleep in seven *Drosophila* species. a** Partial evolutionary tree of the *Drosophila* genus, highlighting in bold the seven species used in this study and their ecological niches (right). **b** World map indicating the geographical ancestral range of each of the species employed in this study. **c** Spontaneous, baseline sleep profile of the seven species across the 24 h period in females (left) and males (right). The white-dark bars under this all figures indicate light-dark periods. ZT: zeitgeber or relative time. The legend below the figure indicates the color code used through the rest of the paper. **d** power analysis of the circadian period of activity in constant dark conditions in all seven species. The shaded area in (**c**) and (**d**) indicate the bootstrapped 95% confidence intervals. **e** Quantification of sleep amount during the day (ZT 0–12, sun icon) and during the night (ZT 12–24, moon icon) in females (left, pink) and males (right, blue) for all seven species. The

pink and cyan rectangles indicate the number of individuals for females and males respectively. The shaded boxes indicate the median and the interquartile range, the whiskers define minimum and maximum values; the dots indicate the single data points. **f** Schematics of the experimental analysis in (**g, h**). Etoscopes collect activity data in real time and deliver air puffs or mock stimuli after random periods of inactivity. A hidden Markov chain model trained on species-specific data classifies sleep stages. **g** The transition parameters of the *D. melanogaster* trained model. **h** The probability of movement following the air puffs (colored boxes) or the mock stimuli (gray boxes) in all seven species, by sleep stage. $N = 46$ male individuals for each species. The shaded boxes indicate the median and the interquartile range; the dots indicate the single data points and define minimum and maximum values.

in their life, being raised and maintained in condition of constant darkness (Supplementary Fig. 7), indicating that lack of sleep rebound after sleep deprivation is not a consequence of circadian masking, but a true reflection of impaired homeostatic control.

### Social interactions vs. mechanical stimuli: differential effects on sleep rebound

While mechanical sleep deprivation is extremely efficient at keeping flies awake (Supplementary Fig. 3a), its ecological relevance is debatable. We therefore explored a more natural way to keep animals awake: male-male interaction in socially naive animals[28,29]. To our surprise, a 24 h sleep deprivation through interaction with a conspecific white-eyed male eventually led to a recognizable rebound in four of the seven tested species (Fig. 2b), all within the closest *melanogaster* subgenus. Still no rebound could be observed in *D. erecta*, *D. willistoni*, *D. virilis*.

The results of these behavioral experiments reveal different evolutionary paths of the two key aspects of sleep regulation in Drosophilids: tightly conserved circadian control, but divergent homeostatic regulation. Understanding the molecular underpinnings of these differences can help shed light not just on sleep evolution, but on the mystery of sleep homeostasis too. Work conducted in the past decades, has consolidated an emerging picture linking synaptic plasticity, learning, and experience to sleep, correlating synaptic strength to sleep needs[30–35]. We[31] and others[32,34,36–40] previously showed that, in *D. melanogaster*, prolonged wake is associated with a reinforcement in synaptic strength conveniently recognizable through an increase in the expression of the synaptic scaffolding protein Bruchpilot (BRP). To check whether this mechanism is common to other *Drosophila* species, we quantified the expression of BRP protein in the brains of the flies after a full day of prolonged wakefulness after mechanical (Fig. 2d) or social sleep deprivation (Fig. 2e). At least for social sleep deprivation, we found a one-to-one correspondence between appearance of rebound sleep and increase in detectable BRP levels (Fig. 2e). For mechanical sleep deprivation, the correspondence was less clear, with *D. simulans* and *D. sechelia* showing an increase in BRP expression without a concomitant manifestation of sleep rebound (Fig. 2d). The other four species, however, showed no increase in rebound and no increase in BRP expression, thus confirming and reinforcing the generality of the previously postulated link between synaptic strength and sleep homeostasis[31,36,40].

### Genetic manipulation unveils the role of synaptic strength in sleep homeostasis

Hypothesizing this link could be more than just correlative, we reasoned it should be possible to alter *D. melanogaster* sleep homeostasis by interfering with its synaptic regulatory machinery. In other words, by reducing synaptic strength in *D. melanogaster*, we may also reduce sleep rebound after mechanical sleep deprivation, as observed in other Drosophilids. To this end, we performed a targeted genetic manipulation using RNAi[41] to pan-neuronally knock-down a selected panel of

seven genes known to be involved in synaptic plasticity (Fig. 3a). We found that, out of these seven genes, four led to a phenocopy of the non-melanogaster species when knocked-down in the brain, that is: a partial or total reduction of homeostatic rebound after mechanical sleep deprivation, but a significant rebound after male-male sleep deprivation (Fig. 3a). These were: the cAMP phosphodiesterase *dunce*[42], the vesicle regulator *synapsin*[43], the regulator of synaptic morphogenesis *starrynight*[44], and the *Drosophila* analogue of the β-amyloid protein precursor *appl*[45]. The translational regulator *orb2*[46] dramatically affected all sleep to an extent that differences in rebound could not be properly assessed and compared. RNAi knock-down of those four genes either altered rebound after mechanical sleep deprivation without decreasing homeostatic rebound after male-male interaction, showing a behavioral phenotype analogous to the one described in *D. simulans, D. sechellia*, or *D. yakuba* (Fig. 3a). A similar effect was also observed in constitutive *synapsin* knock-out mutants (Supplementary Fig. 8b).

### Identifying neuronal pathways influencing sleep rebound

The RNAi knock-down phenotype conveniently opened the perspective of finding the neurons responsible for such a regulation in *D. melanogaster*, and we therefore performed a second small targeted RNAi screen looking for known sleep-regulating populations that would interfere with sleep homeostasis after sleep deprivation (Fig. 3b, c). Knock-down of *dunce* or *synapsin* in the dopaminergic PAM neurons (driven by R58E02-GAL4) and knock down of *dunce* in the dorsal fan-shaped body neurons (driven by R72G06-GAL4) interfered with the flies' natural ability to rebound after mechanical sleep deprivation (Fig. 3b, c). Both regions are well known for their role in learning and memory and have been implicated with many aspects of sleep regulation[47–50]. These behavioral and cell-biological differences between *D. melanogaster* and other six species of the same genus suggest that spontaneous (i.e. circadian) sleep and homeostatic sleep evolved independently under different pressures. If this is the case, one can hypothesize that not just cellular underpinnings, but also neurochemical regulators may have followed divergent evolution. To test this last prediction, we fed flies of all the seven tested species with chemicals acting on the two best studied neurotransmitter pathways involved in sleep regulation: chlordimeform (an octopamine[51] agonist, Fig. 4a) and L-dopa (a dopamine[52] precursor, Fig. 4b). Feeding flies with chlordimeform had a wake-promoting effect in all the tested species except *D. yakuba*, with the strongest arousing effect being found in *D. melanogaster, D. sechellia* and *D. virilis*. L-dopa, on the other hand, showed a strong wake-promoting effect in *D. melanogaster* only, with even an opposite somniferous effect in *D. sechellia* and *D. willistoni*. In Drosophilids, the neuro-regulatory dynamics of sleep have also taken divergent evolutionary paths.

### Discussion

Understanding the evolution of sleep in closely related species can provide unique insights into its function and regulation. One of the

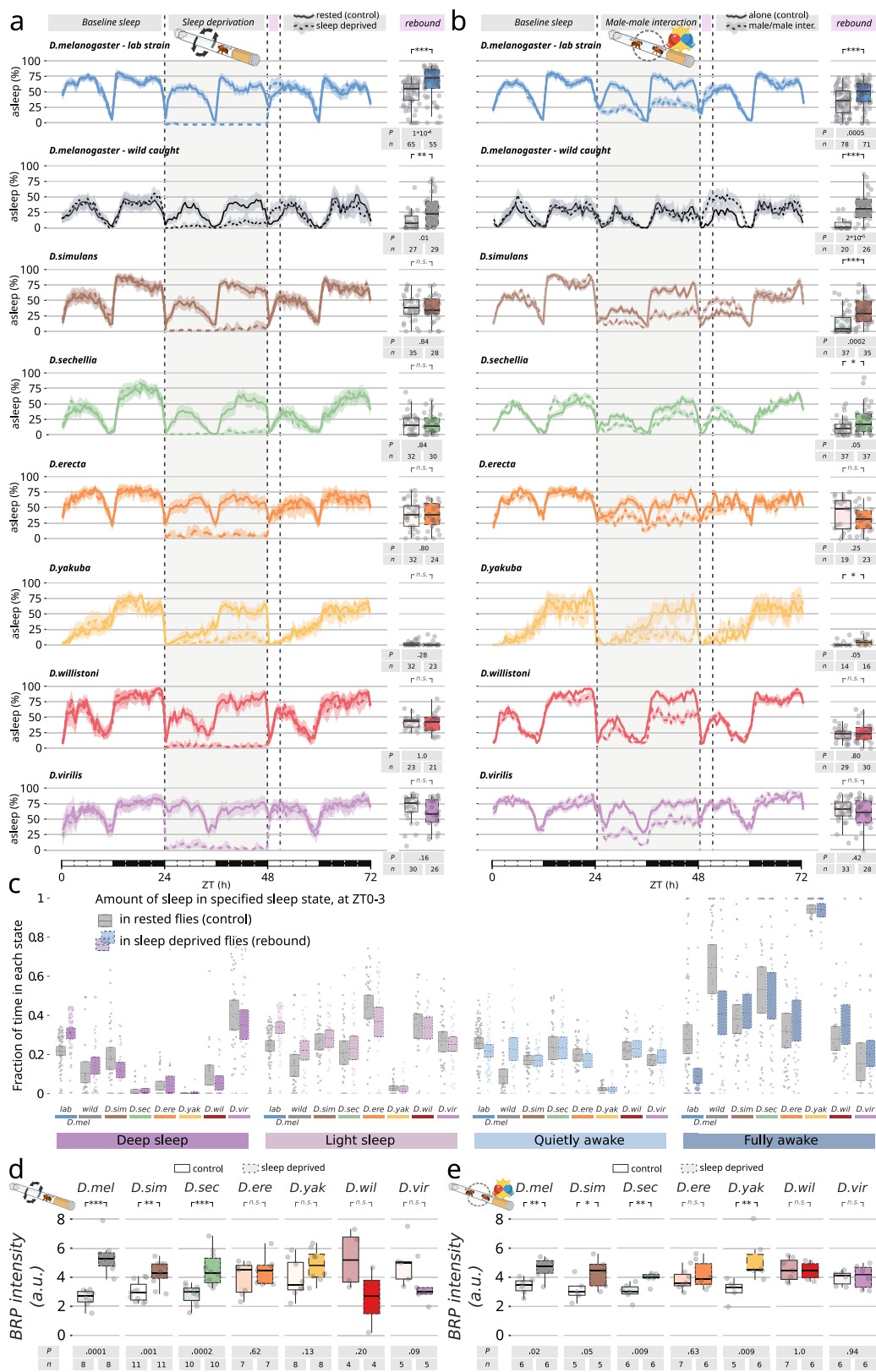

most elegant and powerful models used so far for this goal, is the fish *Astyanax mexicanus* which presents itself in two large conspecific populations: a long-sleeping, eyed population living in the surface of the waters, and several short-sleeping blind populations that were trapped and evolved in the darkness of Mexican caves about 2–5 million years ago, branching in an environment that provided little

or no light-based circadian *zeitgeber* and limited nutrients[1,53]. Here we introduced another powerful framework to study sleep evolution, featuring seven species of *Drosophila* spanning evolutionary distances ranging between 3 and 50 million years. To our surprise, we found that many homeostatic aspects of sleep regulation are not universally shared among these species and are, in fact, a

**Fig. 2 | Different homeostatic rebound after mechanical or social sleep deprivation.** 72 h sleep profile (left) and quantification of rebound (right) in flies subjected to 24 h of mechanically-induced sleep deprivation (**a**), or 24 h of socially-induced sleep deprivation (**b**). Each panel features one strain from each of the seven wild-caught species, or CantonS. In all panels, the sleep profile of rested flies is shown as a continuous line, while sleep-deprived animals are shown in a dashed line. The sleep rebound at ZT 0–3 is quantified on the right side of each sleep profile. Numbers of animals (Ns) and *p* values of sleep-deprived vs. control are shown below each panel. Pairwise comparisons were conducted using the exact version of the Wilcoxon rank-sum test with false discovery rate (FDR) adjustment. The shaded areas indicate the bootstrapped 95% confidence intervals. **c** sleep stage analysis of all species during rebound time (ZT 0–3) after mechanical sleep

deprivation. Increase in sleep depth is observed only in the laboratory strain of *D. melanogaster*. Ns are *D. mel^(lab)*: 200; *D. mel^(wild)* 78; *D. sim* 76; *D. sec* 72; *D. ere* 74; *D. yak* 69; *D. wil* 60; *D. vir* 72. The shaded boxes indicate the median and the interquartile range; the dots indicate the single data points and define minimum and maximum values. Quantification of the change in BRP expression in the brain of sleep-deprived flies after mechanical (**d**) or social male-male (**e**) sleep deprivation. Numbers of animals (Ns) and *p* values of sleep-deprived vs. control are shown below each panel in (**d**) and (**e**). ***$p < 0.001$; **$p < 0.01$; *$p < 0.05$. The shaded boxes indicate the median and the interquartile range, the whiskers define minimum and maximum values; the dots indicate the single data points. Pairwise comparisons were conducted using the exact version of the Wilcoxon rank-sum test with false discovery rate (FDR) adjustment.

prerogative of *D. melanogaster* only (Fig. 4c). Some (*D. simulans, D. sechellia, D. yakuba*) showed signs of sleep homeostasis only following social sleep deprivation and not after mechanical sleep deprivation, while others (*D. erecta, D. willistoni, D. virilis*) never showed any sign of homeostasis in any condition (Fig. 2a–c). Sleep homeostasis is considered a defining hallmark of sleep and if we were to follow the textbook definitions[2], we could reasonably conclude that animals showing no signs of rebound are animals that do not actually sleep: they rest, modulating their inactivity on a circadian basis to avoid danger or maximize the return on investment of their energy expenditure. By this logic, we could argue that *D. erecta, D. willistoni, D. virilis* are non-sleeping species because they never show any sign of rebound following sleep deprivation (Fig. 2). However, this conclusion would be at odds with the fact that all three clearly manifested two different stages of sleep depth characterized by a different threshold of arousal to stimuli (Fig. 1h and Supplementary Figs. 1 and 2). Ecologically speaking, it would be easy to understand why long periods of mere inactivity could be beneficial to a well-fed animal, especially considering they happen at the riskiest times of the day (hot afternoons and dark nights)[54,55], but why this relatively trivial circadian inactivity would manifest itself with different degrees of arousability? After all, if sleep was just adaptive inactivity, as it has been proposed[55], it would benefit from always having a small arousal threshold to avoid predation. We therefore build on the data presented here to reject the null hypothesis that sleep in those species is just adaptive inactivity. Instead, we conclude that the phenomenon of deep sleep—whatever its functions are—has evolved before its own homeostatic regulation: in other words, some animal species show stages of deep sleep that are not homeostatically regulated after sleep deprivation. Based on our results, this appears to be the case in some Drosophilids, and it may be a representation of a more general rule. Stages with different sleep depth in *D. melanogaster* have now been independently described by many groups[22,56–58] and here we show they are very well conserved among Drosophilids. Notably, we found that even their timing is conserved given that in all tested *Drosophila* species, the deepest sleep is consistently observed in the earlier part of the night—a feature well documented in mammals too. Interestingly, the first part of the night is also the only time during which flies are generally more likely to experience deep sleep than light sleep, suggesting a circadianly regulated function for deep sleep (Supplementary Fig. 1d).

In all species analyzed in this work, the presence of sleep rebound was always accompanied by an increase in the Bruchpilot protein, an established bona-fide marker for synaptic strength (Fig. 2d, e). Moreover, the absence of sleep rebound could be phenocopied in *D. melanogaster* by reducing the expression of genes involved in synaptic plasticity in specific anatomical regions involved with learning or sleep (Fig. 3b). The genetic arsenal of non-melanogaster species is still budding, and these latter manipulations must be limited to *D. melanogaster* for now. Nevertheless, these results suggest that different dynamics of synaptic strength may

explain the unexpected differences in sleep homeostasis observed among species, reinforcing the link between sleep homeostasis and synaptic strength. Importantly, genetic manipulation of synaptic plasticity specifically impaired rebound following mechanical sleep deprivation but never affected rebound following social male-male sleep deprivation (Fig. 3a, b), implying that the two apparently similar sleep rebounds are in fact different processes, controlled by a different machinery and/or circuits, and possibly serving different biological functions. This would conveniently explain why some species (*D. simulans, D. sechelia, D. yakuba*) show cellular and behavioral signs of sleep homeostasis only after social male-male sleep deprivation but not after mechanical sleep deprivation. A more specific interpretation lies in the nature of the treatment itself: forced male-male interaction is likely to mimic an ecologically relevant condition of stress, which in turn has tight connections to sleep itself[59,60]. A similar paradigm, resulting in an overlapping outcome, is also been extensively studied in rodents, under the name of social defeat stress[60,61], and it was recently shown to rely on a separate, specific neuronal circuit indeed[62]. It is tempting to speculate that the rebound sleep observed in *D. simulans, D. sechelia, D. yakuba* is driven by a specific post-stress restorative circuit, exercising a post-stress specific function that might have evolved independently of other sleep functions. In other words, synaptic-homeostasis-driven sleep is different from stress-induced sleep, and it has evolved independently of it. We also show that flies can clearly manifest different stages of sleep depth without necessarily needing any apparent homeostatic control over those, indicating that the well modeled link between sleep depth and sleep homeostasis[63,64] does not apply to all species.

Taken together, the differences in behavior, cell-biology, and neuro-pharmacology described here, imply that the evolutionary driving force for sleep in Drosophilids is not homeostasis, as often hypothesized, but circadian adaptation. We propose that sleep in flies did initially evolve as a phenomenon of adaptive inactivity, to limit activity during the more dangerous or inappropriate hours of the day, restraining flies conscious curiosity when it is too dark or too hot. All the other non-trivial functions of sleep—such as regulation of synaptic strength, learning and memory, recovery from stress, modulation of immune response, etc.—which may or may not be specific to some sleep stages, would have then branched divergently, piggybacking on the circadian drive for inactivity in a species-specific manner (Fig. 4c). This may be the common process of sleep evolution in the animal kingdom.

## Methods
### Fly strains
The following VDRC-RNAi transgenic strains were used in this study: UAS-dunceRNAi (#107967), UAS-synapsinRNAi (#109587), UAS-dFRM1RNAi (#110800), UAS-rutabagaRNAi (#101759), UAS-starrynightRNAi (#107993), UAS-applRNAi (#108312) and UAS-orb2RNAi (#11753). The nSyb-GAL4 and MB010B-GAL4 were gifted to

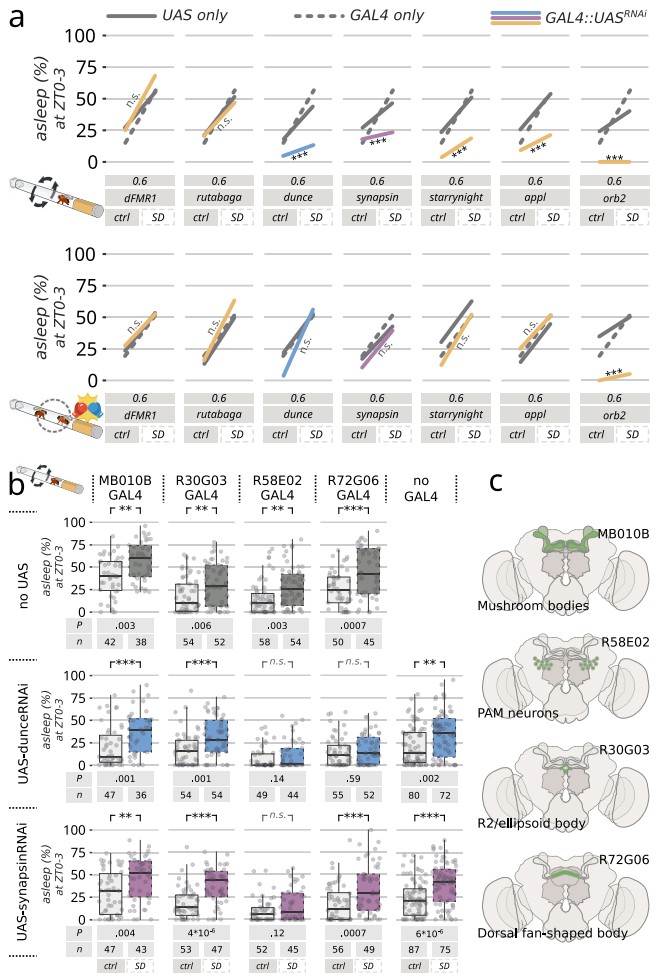

**Fig. 3 | Changes in wake-induced synaptic strength can explain differences among species. a** Changes in rebound sleep at ZT0-3 for flies that underwent RNAi knock-down in one of seven selected genes using the pan-neuronal GAL4 driver nSyb and their relevant parental controls. Flies underwent either 24 h of mechanical sleep deprivation (top) or 24 h of male-male sleep deprivation (bottom). The lines indicate the difference in rebound between control rested conditions (left) and sleep deprivation (right). **b** Rebound sleep at ZT0-3 for flies that underwent RNAi knock-down for *dunce* (blue) or *synapsin* (purple) in five restricted neuronal population driven by four different split-GAL4 drivers. The top-line shows the GAL4 parental control. Through the figure, sleep-deprived flies are shown with a dashed contour while rested control flies are shown in a continuous contour. The shaded boxes indicate the median and the interquartile range, the whiskers define minimum and maximum values; the dots indicate the single data points. *p* values indicated under the box plots were obtained through a two-way ANOVA with Bonferroni correction. **c** Diagram of the expression pattern for the GAL4 lines used in (**b**). Numbers of animals (Ns) and *p* values of sleep-deprived vs. control are shown below each panel. ***$p < 0.001$; **$p < 0.01$; *$p < 0.05$.

Darren Obbard (University of Edinburgh, UK). Two strains of non-melanogaster species were used in Fig. 1c, and the first strain of each were studied in the succeeding figures: *D.simulans* (14021-0251.254 #60, 14021-0251.196 #61), *D.sechellia* (14021-0248.25 #3, 14021-0248.28 #53), *D.erecta* (14021-0224.01 #11), *D.yakuba* (14021-0261.01 #5, 14021-0261.48 #51), *D.willistoni* (14030-0811.24 #1, 14030-0811.13 #55) and *D.virilis* (15010-1051.87 #9, 15010-1051.118 #54). The wildtype species and the white-eyed "intruder" species: *D.simulans w⁻* (14021-0251.133), *D.sechellia w⁻* (14021-02048.30), *D.erecta w⁻* (14021-0248.30), *D.yakuba w⁻* (14021-0261.04), *D.willistoni w⁻* (14030-10811-33) and *D.virilis w⁻* (15010-1051.45) were acquired from The National *Drosophila* Species Stock Center (NDSSC, Cornell University, USA).

## Fly rearing

Flies were raised on standard polenta and yeast based fly media (agar 96 g, polenta 240 g, fructose 960 g, Brewer's Yeast 1200 g in 12 litres of water), referred to as the standard food diet or supplemented with a layer of potato starch (66-117, Nutri-Fly® Instant) (1:4 potato starch:$H_2O$) and filter paper (referred as: the potato starch diet). The species panel were reared on the potato starch diet in all but Fig. 2, where the flies were raised in standard food. The knockdown and mutant data in Fig. 3b–d were raised on the standard food diet. The species food was supplemented with potato starch to improve their viability and filter paper to encourage egg laying and increase progeny number.

## Sleep analysis in ethoscopes

For all experiments, 0–3 days male or virgin female adult flies were sorted into glass tubes (70 × 5 × 3 mm [length × external diameter × internal diameter]) containing standard food. The tubes were loaded into ethoscope sleep arenas (20 animals per device). At least 1 day of baseline was recorded before any treatment. All experiments were carried out under LD conditions unless stated otherwise, 50–70% humidity, in incubators set at 25 °C and with ad libitum access to regular food. Animals that died during the experiment were excluded from the analysis. In order to avoid confounding effects related to the location of the tube on sleep amount (e.g. an ethoscope and incubator), the position of all the tubes was systematically interspersed.

## Circadian analysis in constant darkness

The endogenous period length was established with chi-square $\chi^2$ periodic analysis averaged over 5 days of activity in constant conditions. Flies were both raised and tested with a DD light cycle. For all data presented with a DD light cycle, two approaches were employed: one with brief light exposure during placement of flies into experimental tubes, and another executed in a light-controlled behavioral room where the flies were placed into tubes under red light, and all other light exposure was eliminated. In either case, the flies exhibited slight rhythmicity, and we were unable to eliminate all factors that the flies could potentially entrain to (i.e. temperature and mechanical stimulus); therefore, the data from the two preparations were combined.

## Mechanical sleep deprivation

The effects of mechanical sleep deprivation shown in Fig. 2a were tested using the optomotor module[25], programmed with a 30 s immobility trigger. When triggered, a motor rapidly turns (~400 rpm) for 3 s, spinning the tube housing the fly and preventing sleep. In Fig. 3, the motor stimulus had a duration of 1 s. The mechanical sleep deprivation typically lasted 24 h, except in the case of the continuous mechanical sleep deprivation where it was programmed to sleep deprive the fly for 168 h. Response to a reduced stimulus of ~150 rpm (Supplementary Fig. 3a) were tested with the servo module[15]. Animals recovered for 1 day in the ethoscopes to measure rebound. Animals

us by Crystal Vincent (Imperial College London, UK) and Andrew Lin (The University of Sheffield, UK) respectively, while the other GAL4 lines were obtained from Bloomington *Drosophila* Stock Centre (BDSC, Indiana, USA): R30G03-GAL4 (#49646), R52B10-GAL4 (#69657), R58E02-GAL4 (#41347), R72G06-GAL4 (#39792). All RNAi and GAL4 lines were outcrossed for six generations to the $w^{1118}$ background before testing. The dunce1 (#6020), synapsin97 (#29031), OregonR (#5) and $w^{1118}$ (#3605) flies, were also obtained from BDSC; CantonS strain originally from Ralf Stanewsky (Münster University, Germany). The two *D. melanogaster* wild-caught strains were a gift of

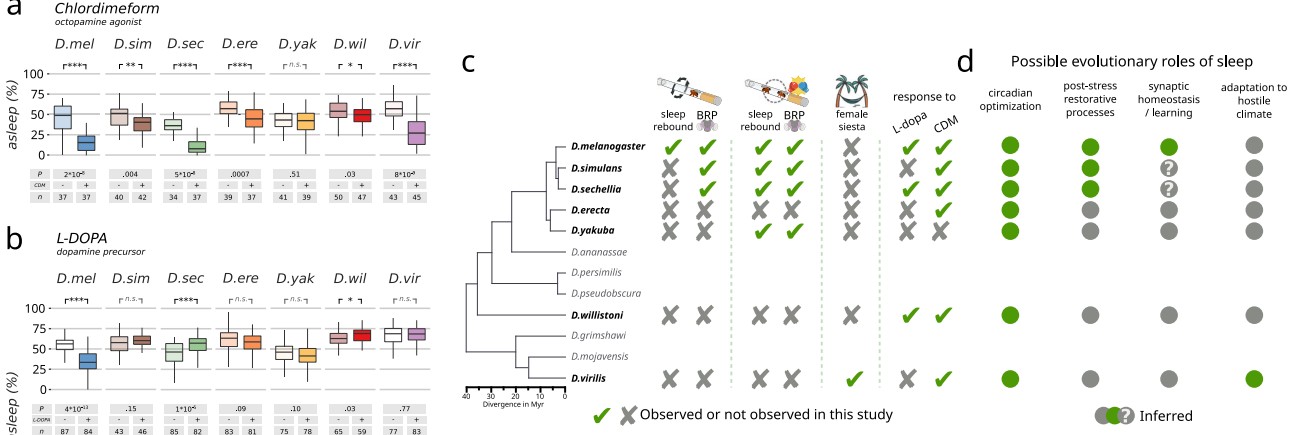

**Fig. 4 | Evolutionary divergence of the neuro-pharmacology of sleep in Drosophilids.** Total sleep over the 24 h period in flies fed with either chlordimeform 0.05 mg/ml (**a**) or L-DOPA 5 mg/ml (**b**). Full colors (right boxes) indicate flies fed with the drugs, while light colors (left boxes) indicate flies fed with vehicle only. Numbers of animals (Ns) and *p* values are shown below each panel. ***$p < 0.001$; **$p < 0.01$; *$p < 0.05$. Pairwise comparisons were conducted using the exact version of the Wilcoxon rank-sum test with false discovery rate (FDR) adjustment. The shaded boxes indicate the median and the interquartile range, the whiskers define minimum and maximum values; single data points are not shown to limit visual complexity. **c** Panoptic summary of the evolutionary divergences found or not found in this work. **d** The proposed model for evolution of different sleep functions, as inferred by the findings in (**c**).

that were asleep >30% of the sleep deprivation period were excluded from the analysis. For the mechanical rebound response in DD (Supplementary Fig. 7), flies were raised and tested in constant darkness as described above with 4 days of baseline, 24 h mechanical sleep deprivation and 1 day recovery for rebound measurement.

### Male-male sleep deprivation
For social interactions, flies were removed from a shared vial and placed in 70 mm × 5 mm glass tubes containing standard food. Twenty tubes were placed in each ethoscope arena. Flies were acclimated in behavioral glass tubes for 4 days of which the last 1–2 days were recorded as a baseline. On the interaction day, intruders (i.e. conspecific white-eyed males) were added at ZT0 and removed in darkness at ZT23-ZT24. All figures show the last baseline day and the first rebound day.

### Analysis of sleep and sleep depth
Arousal threshold during sleep baseline was assessed empirically using the air stimulation module of the ethoscope platform (motorized air/gas/odor delivery module or mAGO)[15]. Flies were placed in an ethoscope at around 4–6 days of age then, after 2 days of baseline sleep recording, ethoscopes were programmed to deliver air puffs after a given period of inactivity ($\tau$) and with a predefined probability factor ($\pi$). Over the several experiments, two different conditions were run: one with an immobility trigger $\tau$ of 30 s and a stimulus delivery probability factor $\pi$ of 0.10, and one with an immobility trigger $\tau$ of 150 s and a stimulus delivery probability $\pi$ of 0.5. This combination of $\tau$ and $\pi$ was chosen because it allowed us to probe sleep episodes of different duration and also to collect a mock dataset that could be used as control to measure spontaneous awakening. After each experiment, behavioral data were analyzed ex post in two ways: (1) fly motion was scored in the 30 s following an air-puffs (or following a mock-stimulus) and each event accordingly classified as awakening or non-awakening; (2) behavior across the 24 h was scored using species-specific hidden Markov chain models in ethoscopy[21]. Those two analyses were then paired to validate if different sleep stages as detected by the HMM model corresponded to a different probability of response after the stimulus. For analysis of sleep states during rebound, the species-specific hidden Markov chain models were applied to the raw activity data from the ethoscopes to investigate the time spent in deep sleep,

light sleep, quietly awake and fully awake after 24 h mechanical sleep deprivation. The HMM model was used in this work only to classify sleep depth (Figs. 1h, 2c and S1d, e, S2a, b). The overall sleep amount in Figs. 1c, e, 2a, b, 3a, b, 4a, b and S1a, S3, S7, S8 was calculated using the 5-min rule.

### Behavioral response to pharmacological stimulation
For the drug treatments, L-dopa (5 mg/ml) was added to sucrose-based food (1% Agar, 5% Sucrose) while the CDM (0.05 mg/ml) was prepared in standard food. One day of baseline behavior was recorded on a control diet (complementary diet minus drug) before flies were flipped onto the drug treatment for 48 h. Successful food consumption was assessed using a blue colored dye[65].

### Immunohistochemistry
The species panel were aged to 10 days in a 25 °C incubator before setting up for 24 h mechanical or social sleep deprivation. At ZT0, on the morning after sleep deprivation, flies were immobilized on ice for brain dissection in 0.1 M phosphate buffered saline (PBS). The brains were fixed in 4% paraformaldehyde for 20 min at room temperature before performing 3 × 10 min washes in PBST (0.3% Triton X in PBS). Samples were then blocked at room temperature for 1 h in 5% normal goat serum (NGS) in PBST followed by incubation with 1:10 mouse anti-nc82 (AB_2314866, DSHB) for 48 h at 4 °C. After primary antibody incubation, the brains were washed for 3 × 10 min in PBST and incubated with 1:200 goat anti-mouse IgG (ab150115, abcam) in 5% NGS for 48 h at 4 °C. Another 3 × 10 min washes in PBST were completed before mounting brains on a microscope slide with Vectashield (Vector Laboratories) for imaging. Brains were imaged with a Leica SP8 - STELLARIS 5 Inverted Light Sheet Confocal Microscope at ×40 magnification to capture z stack images of the dorsal and ventral brain regions. For comparative analysis of the expression levels, images were taken in the same confocal session using identical laser and confocal settings. Analysis of the data was performed using Fiji[66] (NIH, Bethesda). The fan-shaped body (FSB) region was manually delimited stack by stack and a maximum intensity projection was generated. Background measurements were taken from the image background around the brain and subtracted to generate the intensity values. The data was normalized within repeats.

## Statistics

Statistical comparisons were performed as indicated in the text and figure legends, mostly using Wilcoxon rank sum test with false rate discovery (FDR) correction. Survival data were analyzed using pairwise-comparison Log-Rank tests. The analysis of Fig. 3a was done using a two-way anova with Bonferroni correction. In all summary plots, the intermediate reference mark indicates the median and the surrounding error estimates always indicate the bootstrapped 95% confidence intervals. Whenever possible, the entire dataset is shown as a dot plot. Whenever possible, all figures explicitly mention the biological Ns, i.e., the number of biologically independent animals for each data point. Each conclusion relies on multiple independent experiments and never fewer than three independent ones. The actual number of experiments for each panel can be found in the metadata descriptions that are supplied along with the R and Python scripts. Unless differently stated in the legend, all $p$ values arise from Wilcoxon rank sum tests. $p$ values are intended to be supportive and indicate where statistical significance occurs in presence of slight CI limit overlap. In all figures, * are used to indicate customary thresholds of statistical significance: $p < 0.05*$; $p < 0.01**$; $p < 0.001***$. The actual numerical $p$ value is shown in each figure whenever possible and full statistical comparisons among all combinations are available as extended data information in a dedicated file.

## Reporting summary

Further information on research design is available in the Nature Portfolio Reporting Summary linked to this article.

## Data availability

All the metadata describing the experiments and the raw behavioral and confocal data are made publicly available on the data repository Zenodo in a series of four datasets (totaling about 330 Gb) with https://doi.org/10.5281/zenodo.10554851 (https://zenodo.org/records/10554851)[67], https://doi.org/10.5281/zenodo.10557238 (https://zenodo.org/records/10557238), https://doi.org/10.5281/zenodo.10557310 (https://zenodo.org/records/10557310), and https://doi.org/10.5281/zenodo.10966461 (https://zenodo.org/records/10966461). Source data are provided with this paper.

## Code availability

All ethoscope data were analyzed using rethomics[68] or ethoscopy[21]. All the scripts (in R and Python) used to generate the figures in this manuscript as well the related statistical analysis and the original behavioral raw data as obtained with ethoscopes are publicly available through the public dataset repository Zenodo with https://doi.org/10.5281/zenodo.10554851[67]. Software versions used to analyze the data were as follows: behavr: 0.3.2; sleepr: 0.3.0; zeitgebr: 0.3.3; ggetho: 0.3.4; scopr: 0.3.3; ethoscopy 1.3.5.

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

## Acknowledgements

We thank the fly community for sharing reagents and protocols. Special thanks to Darren Obbard for sharing the wild-caught *D. melanogaster* strains. M.J. was supported by EPSRC through doctoral grant EP/R513052/1. F.A.F. was supported by the Imperial President's scholarship. L.B. was supported by BBSRC through doctoral grant BB/M011178/1. A.S.F. was supported by BBSRC through grant BB/R018839/1. L.P.-G.'s laboratory was supported by a European Research Council (ERC) Starting Investigator Grant (802531), a Human Frontiers Science Grant (GY0052/ 2022) and an Allen Institute Distinguished Investigator Award, as well as by The Francis Crick Institute, which is supported by core funding from Cancer Research UK (FC001594), the UK Medical Research Council (FC001594), and the Wellcome Trust (FC001594). We thank the Imperial College London Advanced Hackspace (ICAH) and the Facility for Imaging by Light Microscopy (FILM) at Imperial College London, part-supported by funding from the Wellcome Trust (grant 104931/Z/14/Z) and BBSRC (grant BB/L015129/1).

## Author contributions

M.J., F.A.F., L.B and A.S.F. performed all the experiments and analyzed the data with G.F.G.; M.J., A.S.F., L.P.G. and G.F.G. devised all the experiments. G.F.G. wrote the manuscript and all authors contributed to its editing. A detailed description of authors contributions is provided in Supplementary Table 2.

## Competing interests

The authors declare no competing interests.
