## [Peer Review File · Nature Communications]

Divergent evolution of sleep in *Drosophila* speciesEditorial Note: This manuscript has been previously reviewed at another journal that is not operating a transparent peer review scheme. This document only contains reviewer comments and rebuttal letters for versions considered at Nature Communications. Mentions of the other journal have been redacted.

Reviewer #1 (Remarks to the Author):

This revised manuscript by Joyce et al. titled, "Divergent evolution of sleep functions" uses an evolutionary approach consisting of several *Drosophila* species that span divergent ecological niches to understand whether sleep's evolution is driven by circadian pressures or homeostatic pressures. The group shows convincingly that different *Drosophila* species have all maintained 'circadian-like' rhythms on light:dark settings, yet in only *melanogaster* is the homeostatic pressure maintained. The group reasons that circadian and not homeostatic pressures are driving sleep's conservation. Not to be overlooked, several innovative and powerful approaches support these findings. The group then localized population of synaptic proteins, that when their genes are knocked down using RNAi in *melanogaster* phenocopy the other *Drosophila* sleep patterns, and the authors localize two populations of neurons in particular that are likely mediating these effects: the fan shaped body and PAM DA neurons. Lastly, the group uses pharmacology to show that not just synaptic proteins, but also synaptic transmission is likely involved.

I previously reviewed this manuscript for [redacted] (Then Joyce and French et al) and I thought then that the manuscript was innovative and appeals to a wide audience. My main concerns (circadian regulation on Dark:Dark, arousal threshold in all species, better description of the findings and results) have mostly been addressed. I have two points that I think are noteworthy, but I am very enthusiastic about the work and think it is appropriate for Nature Communications. The approach is innovative, the findings are important, and the study is rigorous.

Here are my two major suggestions. I believe both should be easy to address, and would be valuable not only to the community but also to fortifying the authors claims:

1. The authors use a clever combination of a robotic system developed by the Gilestro lab with a computational pipeline using Hidden Markov Modeling to assess sleep. The authors claim that this liberated their analysis from the typical 5-min definition of sleep, yet no account is given as to what constitutes a sleep like state. Considering the evolutionary perspective of this work, that is something that would be very valuable to the community, and should be easy to include. What are the immobility times required to define a sleep like state in these other species?

2. The authors claim after presenting figure 1 that "aspects of sleep and activity that are meant to be regulated by the circadian clock are generally well conserved." However, as I previously noted even species that do not have a free running clock will still show circadian variations in Light:Dark conditions if they can sense light. The authors did do sleep deprivation in constant darkness (Fig 2 supposed 5). This in itself is an important finding and is interesting, but even here, if I look at the baseline sleep actograms, they look very different. Is sleep a free running rhythm that can be seen in constant darkness.

I would like to see figures 1C and 1D done in D:D conditions. Again, it seems the authors have these data, and they would be central to the claim that circadian pressure, but not homeostatic is conserved. This is essential to make the claim that the 'aspects meant to be regulated by the circadian clock are in tact.' It would also be worth commenting in the text.

2 minor points:

Line 129 - seems to have a missing word: "In line with what observed so far."

Line 168 - "If the different homeostatic regulation observed in non-melanogaster species is connected to different modulation of synaptic strength upon waking...." Could the authors preface this sentence with a sentence stating the basis for this hypothesis. I can guess the logic, but I am not sure how they derived it.

Reviewer #3 (Remarks to the Author):

The manuscript by Joyce et al has improved considerably since its original submission to [redacted]. This is a valuable study that helps move the field in the right direction. While I don't necessarily agree with the overall conclusions about lack of sleep need in certain Drosophilids (see below), the methodology employed to address some important questions is convincing and the results are interesting. The addition of an arousal testing paradigm especially has strengthened the work, along with a clearer view on sleep intensity levels in these fly strains. I have a few suggestions on how the study could be made more accessible to the average reader, as well as couple discussion points that could be worked into the text.

The introduction of sleep intensity metrics and associated levels of arousal (deep sleep, light sleep, quietly awake, fully awake) makes this a much more interesting and believable story. However, the reader remains completely in the dark as to how this was determined. The text and methods suggest a combination of hidden Markov chains and responsiveness to air puffs, alluding to an adaptation of Wiggins 2020 and pointing to a preprint. I suspect most readers will want to see more methodological detail, in this actual paper. There are too many potential variables here (post-hoc Markov analysis of locomotion data in combination with a range of air puff stimuli at potentially different times) to accept the (otherwise very interesting) sleep intensity results at face value. This is novel and still unpublished, so requires better exposition. Even if published elsewhere, the manuscript would still benefit from more explanation here. This is important.

The meaning of sleep functions might need more explanation, especially as it now pertains to potentially different sleep stages. Their finding that deep sleep occurs at the beginning of the night

for most species does seem to suggest a conserved form of sleep homeostatic regulation, which is inconsistent with their conclusions. Also, their finding that most sleep seems to be light (Figure 1, supplement 1e) might also require further discussion, as it calls into question what kind of sleep functions might be subserved by this alternate sleep stage. One possibility is that light sleep and quiet wakefulness are serving similar functions that have been conflated, and by occupying so much of these flies time they were never quite subject to disruption as deep sleep was. If deep sleep functions relate to cellular repair processes, then the absence of homeostatic regulation in some fly species might have a very different explanation (e.g., they might not need to live long) than any effects on light sleep / quiet wakefulness. In other words, the authors fall short of ever really talking about functions, which remains in the title of the manuscript. A deep sleep function might be repair, a light sleep function might be learning. Neither was ever investigated, so I remain perplexed as to what functions are being alluded to, other than the simplistic one about laying low and out of harm's way. I maintain my previous criticism related to this: without having in any way investigated potential cognitive sleep functions, the authors have still not excluded this level of homeostatic regulation in these flies. Including this caveat would make for a more comprehensive discussion.

Many of the figure panels remain very small and hard to appreciate. For example, it is almost impossible to disambiguate the similarly-coloured lines in Figure 1, supplement 1d. Other datasets (e.g. Figure 2c) are packed so densely as to require a magnifying glass (if printed).

While the new agnostic approach to measuring sleep is an excellent idea, it remains unclear where this was applied and where the old 5min criterion remains. For example, were all of the genetic manipulations (Figures 3 & 4) still done the old way, while the new data in Figures 1 and 2 were not? This needs clarification, and rationale if that was indeed the case.

Referees' comments and authors' answers.

Reviewers' text (verbatim with **bold** for emphasis).

Our own replies. Extracts from the manuscripts (verbatim but without references).

Reviewer #1 (Remarks to the Author):

This revised manuscript by Joyce et al. titled, "Divergent evolution of sleep functions" uses an evolutionary approach consisting of several *Drosophila* species that span divergent ecological niches to understand whether sleep's evolution is driven by circadian pressures or homeostatic pressures. The group shows convincingly that different *Drosophila* species have all maintained 'circadian-like' rhythms on light:dark settings, yet in only *melanogaster* is the homeostatic pressure maintained. The group reasons that circadian and not homeostatic pressures are driving sleep's conservation. Not to be overlooked, several innovative and powerful approaches support these findings. The group then localized population of synaptic proteins, that when their genes are knocked down using RNAi in *melanogaster* phenocopy the other *Drosophila* sleep patterns, and the authors localize two populations of neurons in particular that are likely mediating these effects: the fan shaped body and PAM DA neurons. Lastly, the group uses pharmacology to show that not just synaptic proteins, but also synaptic transmission is likely involved. I previously reviewed this manuscript for [redacted] (Then Joyce and French et al) and I thought then that the manuscript was innovative and appeals to a wide audience. My main concerns (circadian regulation on Dark:Dark, arousal threshold in all species, better description of the findings and results) have mostly been addressed. I have two points that I think are noteworthy, but I am very enthusiastic about the work and think it is appropriate for Nature Communications. The approach is innovative, the findings are important, and the study is rigorous.

We thank the reviewer for their very kind words.

Here are my two major suggestions. I believe both should be easy to address, and would be valuable not only to the community but also to fortifying the authors claims:

1. The authors use a clever combination of a robotic system developed by the Gilestro lab with a computational pipeline using Hidden Markov Modeling to assess sleep. The authors claim that this liberated their analysis from the typical 5-min definition of sleep, yet **no account is given as to what constitutes a sleep like state**. Considering the evolutionary perspective of this work, that is something that would be very valuable to the community, and should be easy to include. What are the immobility times required to define a sleep like state in these other species?

This is a very important suggestion, which has been raised by the other reviewer too, and with good reason indeed. We have now added a new supplementary figure in which we described the four types of HMM-predicted bouts, in all species. The figure, reproduced in the next page of this letter for sake of convenience, shows the distribution and characteristics of bout lengths per species. Generally speaking, the HMC model still uses lack of motion to differentiate sleep from wakefulness and mostly uses length limits to separate between sleep bouts. For instance, all the "light-sleep" bouts in *D.sechelia* are shorter than 16 minutes while all of the "deep-sleep" bouts are above 15 minutes. However, the boundary is hardly ever so "sharp". In *D. melanogaster*, for instance, light-sleep bouts can be up to 11 minutes long while deep-sleep bouts can be as short as 2 minutes leaving a large overlap between the two that must be classified depending not on its inherent length, but on the transition probabilities calculated by the model, based on the sequence of bouts preceding it. We have summarised this description at lines 100-105:

"The behavioural bouts identified by the model different by length, with deep sleep and active wake being considerably longer than, respectively, light sleep and quiet wake (Supplementary Fig. 2). However, length is not the sole determinant, given that many episodes of light or deep sleep share the same duration and must therefore be disambiguated on the basis of the model's transition probability matrix (Fig. 1g for *D. melanogaster*, and Supplementary Data for all other species)."

given The new supplementary figure 2, describing the characteristics of the four HMM-defined behavioural states in all examined species.

a, Normalised Kernel density estimation (KDE) of the distribution of behavioural bouts by length in minutes on all species analysed. **b**, Quantitative description of bouts by type. The numbers on each end of the boxplots indicate the minimum (left) and maximum (right) values in minutes for each bout type. The percentages inside each box indicate the fraction of bouts of that state that have lengths overlapping with the adjacent state and are therefore disambiguated by the Hidden Markov Chain model based only on transition probability and not bout length. Percentages on the left indicate the proportion of homogenous bouts, that is bouts that are composed of only inactivity for

sleep or only activity for waking. The vertical bar in each box indicates the median value for that bout.

Another important aspect is that the model prefers bouts characterized by no interruptions of state. In other words, in all species, the totality of sleep bouts (whether light-sleep or deep-sleep) are characterized by total immobility, whereby the “quiet-awake” bouts are characterized mostly by very short bouts of uninterrupted activity. Most of the bouts labelled as “active-awake”, on the other hand, will have prolonged activity occasionally interrupted by short stretches of inactivity. In short, the HMM model cannot offer a rule of thumb that applies to all species, but it does show that prolonged bouts of inactivity are more likely to be associated to deeper sleep.

2. The authors claim after presenting figure 1 that “aspects of sleep and activity that are meant to be regulated by the circadian clock are generally well conserved.” However, as I previously noted even species that do not have a free running clock will still show circadian variations in Light:Dark conditions if they can sense light. The authors did do sleep deprivation in constant darkness (Fig 2 supplement 5). This in itself is an important finding and is interesting, but even here, **if I look at the baseline sleep actograms, they look very different. Is sleep a free running rhythm that can be seen in constant darkness. I would like to see figures 1C and 1D done in D:D conditions.** Again, it seems the authors have these data, and they would be central to the claim that circadian pressure, but not homeostatic is conserved. This is essential to make the claim that the ‘aspects meant to be regulated by the circadian clock are intact.’ It would also be worth commenting in the text.

Figure 1D was actually already done in D:D condition. The conditions of the experiment were only reported in the figure legend and the reader may have missed it, but we have now also added a small graphical addendum to the figure itself, to make it clearer.

As for Figure 2 supp 5 (now known as supplementary figure 7), we agree this is indeed an interesting figure presenting new data also in terms of circadian biology and not just sleep. The experiment was initially done by letting flies develop, emerge, and live in constant D:D, without ever seeing light from the moment the embryo is laid. We think this is the only feasible way to perform this experiment as constant L:L would interfere with normal sleep regulation and would not address the issue of masking that the reviewer initially pointed out. However, as the reviewer noticed, in our first attempt at this experiment, some species still showed some degree of rhythmicity even when the flies were born in total darkness. This may be due to the very short pulse of light they experienced when moved to ethoscopes or possibly to the secondary zeitgeber taking the lead in absence of light, a phenomenon that is well known in circadian biology and studied in *Drosophila* too (see for instance [10.1126/Science.1245710](https://doi.org/10.1126/Science.1245710)). Given these caveats, we have therefore taken some time to repeat this experiment, paying more attention to what we did on the day in which we placed the young flies in the ethoscopes. In particular, on the day when flies were transferred to ethoscope tubes we made sure to operate in total darkness aided only by a red light, and we also scattered the procedure randomly across the 12 hours of the day to avoid any zeitgeber entrainment linked to the physical manipulation of the animals. We reasoned that if all flies are transferred to the ethoscopes at once, they may use this mechanical event to synchronize their clocks. Even by repeating the experiment in this way, some species still show quite a decent rhythmicity which may either be due to secondary zeitgeber that are beyond our control (smell? humidity? mechanical vibrations?) or maternal contribution at the moment of egg-laying. Absolutely interesting, but solving this issue goes beyond the scope of this work. The bottom line is that D:D conditions can make some if not most species arrhythmic while still showing no rebound after mechanical SD (in fact, they show quite considerable negative-rebound!) and this is sufficient to exclude the role of circadian masking, which is the important aspect in terms of our current work.

2 minor points:

Line 129 - seems to have a missing word: “In line with what observed so far.”

We simplified that sentence in the following way L127: “After sleep deprivation, *D. melanogaster* slept longer (and lab raised CantonS *D. melanogaster* slept deeper too – Fig. 2c) but all other species showed no signs of homeostatic rebound. Even more impressively, this difference between species persisted when sleep deprivation was pushed further to cover an uninterrupted period of 168 hours: after seven full days of continuous mechanical sleep deprivation, *D. melanogaster* showed a homeostatic rebound as previously reported²⁹, but none of the other six *Drosophilae* did (Supplementary Fig. 4).”

Line 168 - “If the different homeostatic regulation observed in non-melanogaster species is connected to different modulation of synaptic strength upon waking...” Could the authors preface this sentence with a sentence stating the basis for this hypothesis. I can guess the logic, but I am not sure how they derived it.

That sentence was indeed a bit too convoluted. We have made our assumption more explicit and rephrased it in the following way: (L173)

“Hypothesizing this link could be more than just correlative, we reasoned it should be possible to alter *D. melanogaster* sleep homeostasis by interfering with its synaptic regulatory machinery. In other words, by reducing synaptic strength in *D. melanogaster*, we may also reduce sleep rebound after mechanical sleep deprivation, as observed in other *Drosophilids*. To this end, we performed a targeted genetic manipulation using RNAi⁴⁴ to pan-neuronally knock-down a selected panel of seven genes known to be involved in synaptic plasticity (Fig. 3a).”

Reviewer #3 (Remarks to the Author):

The manuscript by Joyce et al has improved considerably since its original submission to [redacted]. This is a valuable study that helps move the field in the right direction. While I don't necessarily agree with the overall conclusions about lack of sleep need in certain *Drosophilids* (see below), the methodology employed to address some important questions is convincing and the results are interesting. The addition of an arousal testing paradigm especially has strengthened the work, along with a clearer view on sleep intensity levels in these fly strains.

We thank the reviewer for their constructive support.

I have a few suggestions on how the study could be made more accessible to the average reader, as well as couple discussion points that could be worked into the text. The introduction of sleep intensity metrics and associated levels of arousal (deep sleep, light sleep, quietly awake, fully awake) makes this a much more interesting and believable story. However, **the reader remains completely in the dark as to how this was determined. The text and methods suggest a combination of hidden Markov chains and responsiveness to air puffs, alluding to an adaptation of Wiggins 2020 and pointing to a preprint. I suspect most readers will want to see more methodological detail, in this actual paper.** There are too many potential variables here (post-hoc Markov analysis of locomotion data in combination with a range of air puff stimuli at potentially different times) to accept the (otherwise very interesting) sleep intensity results at face value. This is novel and still unpublished, so requires better exposition. Even if published elsewhere, the manuscript would still benefit from more explanation here. This is important.

Both reviewers recommended adding more details on the classification of sleep bouts using HMM, and we therefore added a whole new supplementary figure detailing the findings (please see our response to reviewer #1). In the meantime, the preprint in which we detail the software used for this classification underwent peer-review and was published in September 2023 (

<https://doi.org/10.1093/bioadv/vbad132>, reference 25 on the manuscript). Following the advice of this reviewer, we are now also expanding the methods section to provide more details on how the experiment was conducted (L402):

“Analysis of sleep depth

Arousal threshold during sleep baseline was assessed empirically using the air stimulation module of the ethoscope platform (motorized air/gas/odour delivery module or mAGO)³. Flies were placed in an ethoscope at around 4-6 days of age then, after two days of baseline sleep recording, ethoscopes were programmed to deliver air puffs after a given period of inactivity (τ) and with a predefined probability factor (π). Over the several experiments, two different conditions were run: one with an immobility trigger τ of 30 s and a stimulus delivery probability factor π of 0.10, and one with an immobility trigger τ of 150 s and a stimulus delivery probability π of .50. This combination of τ and π was chosen because it allowed us to probe sleep episodes of different duration and also to collect a mock dataset that could be used as control to measure spontaneous awakening. After each experiment, behavioural data were analysed *ex-post* in two ways: 1) fly motion was scored in the 30 s following an air-puffs (or following a mock-stimulus) and each event accordingly classified as awakening or non-awakening; 2) behaviour across the 24 h was scored using species-specific hidden Markov chain models in ethoscopy⁴. Those two analyses were then paired to validate if different sleep stages as detected by the HMM model corresponded to a different probability of response after the stimulus. For analysis of sleep states during rebound, the species specific hidden Markov chain models were applied to the raw activity data from the ethoscopes to investigate the time spent in deep sleep, light sleep, quietly awake and fully awake after 24hr mechanical sleep deprivation.”

We also rephrased the relevant section in the main text to add detail and context (L77-105):

“To investigate how profound is the evolutionary conservation of sleep in Drosophilids, we created a novel paradigm for the analysis of sleep depth. The system builds²⁵ on a previously described hidden Markov chain (HMC) model⁴⁶, here complemented and validated by a robotic system able to measure arousal threshold by challenging flies with air puffs. These “challenging stimuli” are delivered after random periods of total or partial inactivity (ranging from 1 to 60 minutes; Fig. 1f-h and Supplementary Fig. 1d,e) and at the end of each experiment, response data are analysed *ex post*²⁵ to determine whether behavioural bouts that are differently labelled by the HMC model are also characterized by a different arousal threshold. Strikingly, the validation proved this to be the case (Fig. 1h). Bouts that were labelled as “light sleep” by the model also happened to have a strong probability of response to challenging stimuli, while bouts labelled as “deep sleep” did not. Unsurprisingly, the strongest probability of response was observed when the stimuli were delivered during bouts that the model would have labelled as “wakefulness” (Fig. 1h). Overall, the combined analysis of sleep staging and arousal threshold proved crucial in three ways: i) it confirmed that the ethoscope based video tracking system could indeed reliably detect sleep in all species, even though it was initially developed for *D. melanogaster*¹⁹; ii) it liberated our analysis from the arbitrary definition of sleep as five minutes of immobility (also known as the five minutes rule²⁶⁻²⁸) by adopting an agnostic criterion instead; iii) it allowed us to quantify and describe the actual composition of different sleep stages during the 24 h, as well as their difference across species (Supplementary Fig. 1 and 2). In particular, the analysis showed that all species experience more light sleep than deep sleep, with an average ratio of 1.5 ± 0.4 (mean \pm SD; excluding the outlier *D. erecta* that has a ratio of 10.2). In all species, deep sleep mostly concentrates in the early hours of the night (ZT 12-15 – Supplementary Fig. 1d) and, exception made for *D. erecta* and *D. virilis*, the early night is also the only moment during which flies are more likely to experience deep rather than light sleep. The behavioural bouts identified by the model differ by length, with deep sleep and active wake being considerably longer than, respectively, light sleep and quiet wake (Supplementary Fig. 2). However, length is not the sole determinant, given that many episodes of

light or deep sleep share the same duration and must therefore be disambiguated on the basis of the model's transition probability matrix (Fig. 1g for *D. melanogaster*, and Supplementary Data for all other species)"

The meaning of sleep functions might need more explanation, especially as it now pertains to potentially different sleep stages. **Their finding that deep sleep occurs at the beginning of the night for most species does seem to suggest a conserved form of sleep homeostatic regulation, which is inconsistent with their conclusions.** Also, **their finding that most sleep seems to be light (Figure 1, supplement 1e) might also require further discussion**, as it calls into question what kind of sleep functions might be subserved by this alternate sleep stage. One possibility is that light sleep and quiet wakefulness are serving similar functions that have been conflated, and by occupying so much of these flies time they were never quite subject to disruption as deep sleep was.

We agree that consistent early-night deep-sleep is an important finding, and we extended the analysis and discussion of this observation. Initially at L97:

"In all species, deep sleep mostly concentrates in the early hours of the night (ZT 12-15 – Supplementary Fig. 1d) and, exception made for *D. erecta* and *D. virilis*, the early night is also the only moment during which flies are more likely to experience deep rather than light sleep."

We also extend the discussion regarding the ratio between light sleep and deep sleep, explicitly noting it and quantifying it in the main text L95:

"In particular, the analysis showed that all species experience more light sleep than deep sleep, with an average ratio of 1.5 ± 0.4 (mean \pm SD; excluding the outlier *D. erecta* that has a ratio of 10.2)."

Finally, we bring all together again in the discussion at line 247:

"Notably, we found that even their timing is conserved given that in all tested *Drosophila* species, the deepest sleep is consistently observed in the earlier part of the night – a feature well documented in mammals too. Interestingly, the first part of the night is also the only time during which flies are generally more likely to experience deep sleep than light sleep, suggesting a circadianly regulated function for deep sleep (Supplementary Fig. 1d)."

Please note that we do not really agree with the interpretation that the presence of early night deep sleep can be taken as a sign of homeostasis. In fact, it is generally well accepted that the timing of sleep phases is under control of the circadian clock, to the point that having a late bedtime is generally discouraged because it will remove most of NREM in humans (See for instance [10.1523/JNEUROSCI.15-05-03526.1995](https://doi.org/10.1523/JNEUROSCI.15-05-03526.1995)).

About the ratio between light and deep sleep, our analysis show that most sleep is indeed light in all species studied, although with a ratio of light-sleep to deep-sleep that can vary quite dramatically between species from about 1.5:1 to 10:1. We do not find this surprising. In adults humans, for instance, only 5% of the total sleep time is spent in the deepest sleep stage N1; 50% is in stage N2; and 20% and 25% are in the less deep sleep stages N3 and REM respectively (AASM Scoring Manual). If we consider N1 and N2 to be the deepest stages of sleep, the figures would be strikingly similar. We could in principle draw this parallel between flies and humans in the discussion to give context to the expected ratio between deep sleep and light sleep, but we prefer not to given that the characterisation of sleep stages in humans is largely electrophysiological while ours is behavioural.

If deep sleep functions relate to cellular repair processes, then the absence of homeostatic

regulation in some fly species might have a very different explanation (e.g., they might not need to live long) than any effects on light sleep / quiet wakefulness. In other words, the authors fall short of ever really talking about functions, which remains in the title of the manuscript. **A deep sleep function might be repair, a light sleep function might be learning. Neither was ever investigated**, so I remain perplexed as to what functions are being alluded to, other than the simplistic one about laying low and out of harm's way. I maintain my previous criticism related to this: without having in any way investigated potential cognitive sleep functions, the authors have still not excluded this level of homeostatic regulation in these flies. Including this caveat would make for a more comprehensive discussion.

We believe all we can reasonably do at this stage is to keep our options open and be explicit about the fact that we do not know what the species-specific functions of sleep are, or how they correlate to specific sleep stages (if at all). Speculating here about cellular repair, learning *etc* would be misplaced and premature. The comment from the reviewer, however, does indicate that we need to be more explicit about what our conclusions are. We have rephrased the last paragraph of the discussion to stress one aspect: we are not proposing that adaptive inactivity is the only function of sleep. In fact quite the opposite! We now state this loudly at line 240:

“We therefore build on the data presented here to reject the null hypothesis that sleep in those species is just adaptive inactivity.”

Instead, we are proposing that adaptive inactivity is the function that sleep has “used” to evolve upon. There may be other crucial, possibly vital functions that are species specific and, in principle, they could even be associated to different sleep stages in different animals. Our closing paragraph (L279) now reads:

“Taken together, the differences in behaviour, cell-biology, and neuro-pharmacology described here, imply that the evolutionary driving force for sleep in Drosophilids is not homeostasis, as often hypothesized, but circadian adaptation. We propose that sleep in flies did initially evolve as a phenomenon of adaptive inactivity, to limit activity during the more dangerous or inappropriate hours of the day, restraining flies conscious curiosity when it is too dark or too hot. All the other non-trivial functions of sleep – such as regulation of synaptic strength, learning and memory, recovery from stress, modulation of immune response, *etc.* – which may or may not be specific to some sleep stages, would have then branched divergently, piggybacking on the circadian drive for inactivity in a species-specific manner (Fig. 4c). This may be the common process of sleep evolution in the animal kingdom.”

Many of the figure panels remain very small and hard to appreciate. For example, it is almost impossible to disambiguate the similarly-coloured lines in Figure 1, supplement 1d. Other datasets (e.g. Figure 2c) are packed so densely as to require a magnifying glass (if printed).

We appreciate this issue and we have the same concern. We have enlarged some of the plots in the smallest panels, and without doubt will revisit the size of each panel once the paper is accepted and a page limit is set. Only then, will we finally know what the exact readability of each figure is. Supplementary figures will be enlarged to cover the entire page.

While the new agnostic approach to measuring sleep is an excellent idea, it remains unclear where this was applied and where the old 5min criterion remains. For example, were all of the genetic manipulations (Figures 3 & 4) still done the old way, while the new data in Figures 1 and 2 were not? This needs clarification, and rationale if that was indeed the case.

This is another excellent point. We now discuss how the Hidden Markov Chain model compares to the five-minutes rule at line 105:

“While providing the unique ability of separating light sleep from deep sleep, the HMC analysis also confirms that the original five minutes rule can still be used to disambiguate sleep from wakefulness given that, across all species, 100% of the bouts of inactivity lasting five or more minutes are classified as sleep by the model. In non-melanogaster species, 100% of the deep sleep episodes feature at least 8 minutes of consecutive inactivity (Supplementary Fig. 2b).”

This implies that for the purpose of measuring mere sleep amount, the HMC model and the five-minutes rule are interchangeable. We now state that explicitly in the materials and methods (L421).

Reviewer #3 (Remarks to the Author):

The authors have addressed most of my concerns. This is an excellent study that will be an important reference for the *Drosophila* sleep field, and for the understanding of the evolution of sleep more generally. While I am happy for this work to be published as soon as possible, I have a few strong recommendations for the authors, as well as a few minor points and suggestions.

Recommendation 1:

The paper starts in a robust manner, by re-defining sleep in flies according to a mixed approach combining Markov modelling and arousal thresholds. This is how different sleep stages are identified and compared among the *Drosophila* species, also after sleep deprivation – providing some very interesting results. The second half of the paper then seems to revert to a simpler sleep measure (% asleep), suggesting a return to the 5-min criterion for this metric. Even though the authors addressed this point in their rebuttal, it remains quite unclear whether % asleep follows the 5-minute rule. This is important because the authors themselves show (in their earlier figures) that the 5min rule might not be an appropriate way to compare species – or even mutants for that matter. This potential point of confusion might not change their conclusions, but they do need to be explicit in saying which sleep datasets followed the 5min rule, e.g. in the figure legends. To this reviewer, it seems that the 5min rule was applied to Figure 1c,e; Figure 2a,b; Figure 3a,b; Figure 4a,b. Is this correct? If so, this should be pointed out and perhaps some question could be raised about whether % sleep measured this way is comparable between species, following this *melanogaster*-derived criterion. Readers might wonder why revert to this simple metric after having shown sleep intensity differences. I am not asking for new analysis, just clarification – in the manuscript.

Recommendation 2:

Although the authors justifiably remain agnostic about sleep functions in their discussion, they are taking a strong position about functions in the manuscript's title. This could be confusing to some readers, especially those outside the fly sleep field. Fly movement is not the only function that could be homeostatically regulated and thus detected by a rebound following sleep deprivation. Using the same measure (movement) to identify a sleep function (or lack thereof) as is used to identify sleep (lack of movement) presents possibly circular logic that may be difficult to disentangle. By relegating sleep functions to movement alone, the authors arrive at a conclusion which remains highly debatable: that adaptive inactivity was the first (evolutionary) sleep function, with other functions (learning, synaptic homeostasis, stress regulation, etc) 'piggybacking' on these circadian-driven periods of inactivity. While an interesting idea worth proposing and testing, this seems unlikely. It is for example well-accepted that even nematodes display sleep-like functions, and these are associated with developmental processes wherein this otherwise never-resting animal stops moving to accomplish important growth / rewiring processes during development. This is evolutionarily ancient and has nothing to do with adaptive inactivity, yet aspects of nematode sleep are recapitulated in the deep sleep of other animals (e.g., cell growth and rewiring). While it is understandable that investigating these diverse sleep functions is beyond the scope of this study, the implication from the title is that sleep functions have been assessed. The study was on mobility alone as a measured sleep function, and BRP expression to some extent. I'm not sure this justifies the current title, or supports the strong conclusion made at the end about adaptive inactivity being key. I should emphasize that in no way am I suggesting that this diminishes an otherwise fantastic study. I just propose the authors be cautious in their conclusions, and maybe rethink their title. A more appropriate title might be "Divergent evolution of sleep in *Drosophila* species".

Other suggestions / questions:

Line 166: The authors describe a 'one-to-one correlation'. Perhaps 'correlation isn't the best term here, and 'correspondence' might be more appropriate?

Line 168: 'slightly more spurious' could be replaced with 'the correspondence was less clear'?

The panoptic summary in Figure 4C could be confusing, especially for the grey portion. Does an X mean that there is no data, or that this does not exist in said species?

Thanks for providing this final round of welcome suggestion. Please see our response and actions below.

REVIEWERS' COMMENTS

Reviewer #3 (Remarks to the Author):

The authors have addressed most of my concerns. This is an excellent study that will be an important reference for the Drosophila sleep field, and for the understanding of the evolution of sleep more generally. While I am happy for this work to be published as soon as possible, I have a few strong recommendations for the authors, as well as a few minor points and suggestions.

Recommendation 1:

The paper starts in a robust manner, by re-defining sleep in flies according to a mixed approach combining Markov modelling and arousal thresholds. This is how different sleep stages are identified and compared among the Drosophila species, also after sleep deprivation – providing some very interesting results. The second half of the paper then seems to revert to a simpler sleep measure (% asleep), suggesting a return to the 5-min criterion for this metric. Even though the authors addressed this point in their rebuttal, it remains quite unclear whether % asleep follows the 5-minute rule. This is important because the authors themselves show (in their earlier figures) that the 5min rule might not be an appropriate way to compare species – or even mutants for that matter. This potential point of confusion might not change their conclusions, but they do need to be explicit in saying which sleep datasets followed the 5min rule, e.g. in the figure legends. To this reviewer, it seems that the 5min rule was applied to Figure 1c,e; Figure 2a,b; Figure 3a,b; Figure 4a,b. Is this correct? If so, this should be pointed out and perhaps some question could be raised about whether % sleep measured this way is comparable between species, following this melanogaster-derived criterion. Readers might wonder why revert to this simple metric after having shown sleep intensity differences. **I am not asking for new analysis, just clarification – in the manuscript.**

We actually had a line in the materials and methods explaining when the model was used. The line read “The HMM model was used in this work only to classify sleep depth. Overall sleep amount was still calculated using the five-minutes rule”. However, we agree that this could be made more explicit, so we have now rephrased that to read as follows (L393): “The HMM model was used in this work only to classify sleep depth (Fig. 1h; 2c; S1d,e; S2a,b). The overall sleep amount in Fig. 1c,e; 2a,b; 3a,b; 4a,b; S1a, S3; S7; S8; was calculated using the five-minutes rule.”

The rationale for this decision is clearly explained at L117 “While providing the unique ability of separating light sleep from deep sleep, **the HMC analysis also confirms that the original five minutes rule can still be used to disambiguate sleep from wakefulness** given that, across all species, 100% of the bouts of inactivity lasting five or more minutes are classified as sleep by the model.”

Recommendation 2:

Although the authors justifiably remain agnostic about sleep functions in their discussion, they are taking a strong position about functions in the manuscript’s title. This could be confusing to some readers, especially those outside the fly sleep field. Fly movement is not the only function that could be homeostatically regulated and thus detected by a rebound following sleep deprivation. Using the same measure (movement) to identify a sleep function(or lack thereof) as is used to identify sleep (lack of movement) presents possibly circular logic that may be difficult to disentangle. By relegating sleep functions to movement alone, the authors arrive at a conclusion which remains highly debatable: that adaptive inactivity was the first (evolutionary) sleep function, with other functions (learning, synaptic homeostasis, stress regulation, etc) ‘piggybacking’ on these circadian-driven periods of inactivity. While an interesting idea worth proposing and testing,

this seem unlikely. It is for example well-accepted that even nematodes display sleep-like functions, and these are associated with developmental processes wherein this otherwise never-resting animal stops moving to accomplish important growth / rewiring processes during development. This is evolutionarily ancient and has nothing to do with adaptive inactivity, yet aspects of nematode sleep are recapitulated in the deep sleep of other animals (e.g., cell growth and rewiring). **While it is understandable that investigating these diverse sleep functions is beyond the scope of this study, the implication from the title is that sleep functions have been assessed.** The study was on mobility alone as a measured sleep function, and BRP expression to some extent. I'm not sure this justifies the current title, or supports the strong conclusion made at the end about adaptive inactivity being key. I should emphasize that in no way am I suggesting that this diminishes an otherwise fantastic study. **I just propose the authors be cautious in their conclusions, and maybe rethink their title. A more appropriate title might be "Divergent evolution of sleep in *Drosophila* species".**

The reviewer makes a sensible and fair point: the manuscript does not really, empirically focus on sleep functions, and it would be better to pick a title that more accurately reflects the work's findings. We stand convinced and changed the title of the paper exactly as the reviewer suggested: "**Divergent evolution of sleep in *Drosophila* species**". We chose to keep these concepts in the discussion however because we believe they add intellectual value to our findings, and they are unambiguously separated from the results section.

Other suggestions / questions:

Line 166: The authors describe a 'one-to-one correlation'. Perhaps 'correlation isn't the best term here, and 'correspondence' might be more appropriate?

Good suggestion, thanks. Done (L178).

Line 168: 'slightly more spurious' could be replaced with 'the correspondence was less clear'?

Done too (L180).

The panoptic summary in Figure 4C could be confusing, especially for the grey portion. Does an X mean that there is no data, or that this does not exist in said species?

The reviewer is correct. Upon their suggestion, we have redrawn that model in a more explicit way (see below).

Reviewer #1 (as prompted by the editor in the checklist):

Thank you for clarifying that Figure 1d is for D:D individuals. However, Reviewer #1 also asked for the plots corresponding to Figure 1c for D:D individuals. These can be included in the Supplementary Information.

We have now added a third panel to Supplementary Figure 7:

c, Overlapping plot of baseline sleep in constant DD conditions for all seven species. Data are extracted from the dataset in b and highlight that some limited rhythmicity remains in some of the species (e.g. *D. willistoni* and *D. sechelia*) even when these were developed and raised in total darkness as mentioned in the methods (Re: Circadian analysis in constant darkness)